# Iron Metabolism following Twice a Day Endurance Exercise in Female Long-Distance Runners

**DOI:** 10.3390/nu14091907

**Published:** 2022-05-02

**Authors:** Aya Ishibashi, Naho Maeda, Chihiro Kojima, Kazushige Goto

**Affiliations:** 1Department of Life Sciences, Graduate School of Arts and Sciences, The University of Tokyo, Tokyo 153-8902, Japan; aya-ishibashi@g.ecc.u-tokyo.ac.jp; 2Graduate School of Sport and Health Science, Ritsumeikan University, Kusatsu 525-8577, Japan; maeda.nh0709@gmail.com; 3Department of Sports Science, Japan Institute of Sports Science, Tokyo 115-0056, Japan; chihiro.kojima@jpnsport.go.jp

**Keywords:** hepcidin, iron deficiency, endurance exercise, female athlete

## Abstract

Iron deficiency anemia (IDA) and iron deficiency (ID) are frequently observed among endurance athletes. The iron regulatory hormone hepcidin may be involved in IDA and/or ID. Endurance athletes incorporate multiple training sessions, but the influence of repeated bouts of endurance exercise within the same day on iron metabolism remains unclear. Therefore, the purpose of the present study was to investigate the influence of twice a day endurance exercise on iron metabolism, including the hepcidin level, in female long-distance runners. Thirteen female long-distance runners participated in this study. They completed the twice-a-day endurance exercise in the morning and afternoon. Blood samples were collected four times in total: at 06:00 (P0), 14:00 (P8), 20:00 (P14), and 06:00 the next day (P24). In addition to the blood variables, nutritional intake was assessed throughout the exercise day. Serum hepcidin levels were significantly elevated (compared to P0) until the following morning (P24). Moreover, dietary analysis revealed that subjects consumed a low volume of carbohydrates (<6 g/kg body mass/day). In conclusion, twice a day endurance exercise resulted in significant elevation of serum hepcidin level 24 h after completion of the exercise in female long-distance runners. Therefore, athletes with a high risk of anemia should pay attention to training frequency and nutritional intake in order to maintain optimal iron metabolism.

## 1. Introduction

Iron deficiency anemia (IDA) is frequently observed among endurance athletes [1]. Previous studies reported that a portion of elite athletes are regarded as iron deficient (ID) [2,3]. A low ferritin level with ID attenuates endurance performance even in the absence of IDA [4]. In addition, once athletes are diagnosed with IDA, it takes about 3 months to recover [5]. Therefore, prevention of IDA and/or ID is essential to improve endurance performance.

Hepcidin is a crucial mediator of iron homeostasis, and it is believed to be associated with exercise-induced ID. The serum hepcidin level was transiently elevated at ~3 h after completion of the exercise [6,7,8]. Moreover, exercise-induced hepcidin elevation was associated with increases in exercise-induced inflammation and hemolysis [9]. Strenuous exercise promotes inflammation, as reflected by a marked increase in the plasma interleukin-6 (IL-6) level following exercise [10]. Pro-inflammatory cytokines, such as IL-6, stimulate hepcidin production, and sustained inflammation induced by consecutive days of endurance training may promote hepcidin production and ID during the training period [11,12,13].

In a laboratory setting, most of the previous studies demonstrated that a single bout of endurance exercise increased hepcidin level post-exercise. However, in practice, athletes frequently take part in multiple training sessions per day (up to two or three times a day) in some endurance events (e.g., long-distance runners, triathletes). Ronsen et al. [14] investigated that effect of twice high-intensity endurance exercise (cycling) vs. a single bout of exercise session on IL-6. Consequently, the second bout of exercise on the same day caused a more pronounced increase in IL-6 compared with the first bout of exercise in elite athletes. Muscle glycogen content had not fully recovered from the previous bout of endurance exercise before undergoing the second bout on the same day [15]. This may induce an energy crisis in working muscles, affecting both carbohydrate and fat metabolism. Therefore, exercise-induced hepcidin elevation during the second bout of exercise may be augmented, as shown by facilitated IL-6 response.

According to the international sport nutrition guidelines, 6–12 g/kg BM/day carbohydrate intake is currently recommended for an endurance program (e.g., 1–3 h/day) and/or high-intensity exercise (e.g., 4–5 h/day) [16]. In our study, three consecutive days of endurance training under low energy and carbohydrate (4 g/day) intake reduced the muscle glycogen content, with concomitant increase in baseline serum hepcidin level in male long-distance runners. On the other hand, when sufficient carbohydrate (about 9 g/kg/day) was consumed during the same period, baseline hepcidin did not change significantly throughout the days of the training period [17]. Additionally, carbohydrate ingestion inhibited IL-6 mRNA and plasma cytokine levels after 3 h of running [18], and nutritional interventions attenuated the exercise-induced IL-6 elevation [19]. However, it is unclear how many carbohydrates female long-distance runners consume on actual training days and how much this affects their iron metabolism.

We previously observed that an increase in the monthly running distance significantly augmented the baseline hepcidin level in female long-distance runners [20]. To examine the effects of hepcidin on exercise, the majority of previous studies utilized running on a treadmill [7,21,22] or cycling [23] in the laboratory setting. In recent years, field-based studies targeting athletes have been reported [24], However, somewhat surprisingly, the influence of repeated bouts of endurance exercise within the same day on iron metabolism has not been fully determined under practical situation (e.g., during actual training session on a track). Therefore, we were challenged to collect in-field data in response to actual training sessions on the track.

Thus, in the present study, we investigated the influence of twice a day endurance exercise sessions on iron metabolism, including the hepcidin level in female long-distance runners. We hypothesized that twice a day endurance exercise sessions would increase serum hepcidin levels over the day.

## 2. Materials and Methods

### 2.1. Subjects

In total, 13 female long-distance runners (mean age ± standard error (SE): 20.7 ± 0.4 years) participated in this study. All subjects belonged to the same track and field team at university, and they were classified as being top-class long-distance runners in the university category in Japan. This team has won the All Japan University Women’s Ekiden (i.e., long distance running multi-stage relay race) for five consecutive years. Additionally, they lived in the same dormitory, exclusive to the long-distance running team. They regularly participated in endurance exercise twice a day (1 h in the early morning, 2 h in the afternoon) for 6 days per week. None of the subjects were using iron replacement. The subjects were informed about the study’s purpose and procedures, and they provided written informed consent. This study was approved by the Ethics Committee for Human Experiments at the Japan Institute of Sports Sciences (IRB-2016-033), Japan. The study was performed in compliance with the Declaration of Helsinki.

### 2.2. Experimental Design

None of the subjects trained the day before the experiment. As shown in Figure 1, the exercise consisted of two sessions: one in the morning (7:00 to 8:00; 1 h) and one in the afternoon (15:00 to 17:00; 2 h). Blood samples were collected four times in total, at 06:00 (P0; before exercise following an overnight fast), 14:00 (P8; 8 h after completing the morning exercise), 20:00 (P14; 3 h after completing the afternoon exercise), and 06:00 the next day (P24; following an overnight fast). In addition to the blood variables, we also assessed body composition and nutritional intake throughout the day of exercise.

### 2.3. Measurements

#### 2.3.1. Body Composition

Height was measured using a height meter (ST-2M, Yagami Co., Aichi, Japan). Body mass and fat mass were evaluated using a multi-frequency impedance technique (InBody 720, Biospace, Seoul, Korea). The InBody 720 accurately measures the body’s water content and body composition, including fat, free fat, and skeletal muscle mass, using a range of frequencies from 1 kHz to 1 MHz [25]. The body composition was evaluated before breakfast. For the measurement prior to the impedance, subjects emptied their bladders before and wiped the soles of their bare feet with alcohol cotton. Then, sex, age, and height were entered directly into the instrument. Subject’s bare feet were placed on the metal plates of the scale and the subject firmly grasped the hand grips while placing the thumb and fingers in the standard location, as indicated in the operation manual. The InBody 720 calculates 30 impedance measurements by using 6 different frequencies (1, 5, 50, 250, 500, and 1000 kHz) at each of 5 different segments, which include the right arm, left arm, trunk, right leg, and left leg.

#### 2.3.2. Blood Sampling and Analyses

Blood samples were collected at P0, P8, P14, and P24. All blood samples were obtained via an antecubital vein while in a seated position. After drawing blood, serum and plasma samples were obtained by centrifugation (3000 rpm, 10 min, 4 °C) and stored at −80 °C until analysis.

Blood hemoglobin (Hb) levels were evaluated at a clinical laboratory (Falco Holdings, Kyoto, Japan). Serum iron, ferritin, and haptoglobin levels were also evaluated at a clinical laboratory (SRL, Tokyo, Japan). Plasma IL-6 and serum hepcidin levels were determined by enzyme-linked immunosorbent assay using commercially available kits (R&D Systems Inc., Minneapolis, MN, USA). All samples were analyzed in duplicate, and the average values were determined. The intraassay coefficients of variation were 5.0% (IL-6) and 4.4% (hepcidin).

#### 2.3.3. Nutritional Assessment

We also conducted a dietary survey using food diaries and provided verbal and written instructions to ensure accurate recording of all foods and fluids consumed. The subjects were also requested to take photos and weigh their plates before and after a meal. Once the subjects submitted their dietary surveys, a dietitian checked the food records and confirmed the contents, clarifying specific items and/or detailed information, as necessary. Dietary analysis was conducted using specially designed software (Eiyo-kun, Kenpaku-sha, Tokyo, Japan). Information about using nutrient supplements was also collected during the recall visits. Nutrient intake from supplements (if any) was included in the data.

#### 2.3.4. Statistical Analyses

All data are presented as means ± SE. The normality of the data distribution was assessed by the Shapiro–Wilk test. Changes in blood variables over time were evaluated by one-way analysis of variance (ANOVA) with repeated measures time (P0, P8, P14, and P24). When the ANOVA revealed a significant interaction or a main effect, Tukey’s post-hoc analysis was performed to explore this difference further. A paired *t*-test was used to compare variables between nutritional intake between on the day before exercise and the day of exercise. In addition to *p*-values, we calculated Cohen’s d-values (on independent *t*-test) or the partial η^2^ values (when the ANOVA with repeated measures was performed) to determine effect size (ES). All analyses were performed using SPSS software ver. 22.0 (SPSS Inc, Chicago, IL, USA). A *p*-value < 0.05 was considered to indicate statistical significance.

## 3. Results

### 3.1. General Information

The average height was 160.2 ± 0.9 cm, the average body mass was 47.5 ± 0.7 kg, and the average percentage of body fat was 15.4 ± 0.8%. The average running distance was 22.4 ± 0.9 km/day through the exercise day.

### 3.2. Blood Parameters

#### 3.2.1. Fasting Iron Parameters in the Morning

The blood Hb levels were 12.7 ± 0.1 g/dL (P0) and 12.4 ± 0.4 g/dL (P24) (F = 0.35, *p* = 0.16, and d = −0.75). The serum ferritin levels were 24.3 ±5.4 g/dL (P0) and 25.4 ± 5.8 g/dL (P24) (F = 0.82, *p* = 0.89, and d = 0.06). The blood Hb and serum ferritin levels did not show significant change over time.

#### 3.2.2. Haptoglobin, Iron, IL-6, and Hepcidin Level after Exercise

Figure 2 presented the exercise-induced changes in the blood variables. The serum haptoglobin level (F = 0.08, *p* = 0.97, η^2^ = 0.01) and IL-6 (F = 0.57, *p* = 0.47, η^2^ = 0.01) did not show significant change over time. The serum iron level showed significant main effects for time (F = 0.471, *p* = 0.016, η^2^ = 0.29). The serum iron level was significantly higher at P24 than at P0.

The serum hepcidin level showed significant main effects for time (F = 4.2, *p* = 0.01, η^2^ = 0.26). The serum hepcidin level was significantly higher at P8, P14, and P24 than at P0.

### 3.3. Nutritional Intake during the Exercise Periods

Table 1 shows the nutritional intake on the day before exercise and the day of exercise. Percentage of fat and fat intakes from the daily diet were significantly decreased on the day of exercise. In contrast, % carbohydrate intakes were significantly increased on the day of exercise.

## 4. Discussion

We investigated the serum hepcidin response to twice a day endurance exercise (actual training sessions on the track) in female long-distance runners. We showed that twice a day endurance exercise elevated serum hepcidin level through the following morning. Hepcidin acts to internalize and degrade ferroportin export channels in the small intestine and on the surface of macrophages, resulting in attenuated absorption of dietary iron in the gut and preventing the release of iron from macrophages [12]. Therefore, twice-a-day endurance exercise (a typical training regime for endurance athletes) is likely to have a strong impact on iron metabolism, with increased the risk of exercise-induced ID.

The hepcidin level is elevated after a single bout of endurance exercise and generally returns to baseline within 24 h after the exercise [8]. In the present study, it was notable that the serum hepcidin level remained significantly higher the following morning (24 h after the first bout of endurance exercise). These results may be explained by low muscle glycogen with a lower carbohydrate intake. Reduced muscle glycogen content has been shown to facilitate exercise-induced IL-6 elevation [17]. Ronsen et al. [14] reported that glycogen depletion and muscle damage caused by twice a day of exercise induced an increase in IL-6. It may affect the increase in exercise-induced hepcidin elevation in the present study.

In the international sport nutrition guidelines, preparation for endurance events (< 90 min exercise) requires 7–12 g/kg/24 h for daily fuel [26]. In the present study, carbohydrate intake ranged between 5.3 and 6.0 g/kg BM/day. Thus, insufficient carbohydrate intake and muscle glycogen level may be associated with augmented exercise-induced hepcidin elevation until the following morning. Moreover, the influence of manipulating carbohydrate intake (3 vs. 10 g/kg BM) after a glycogen-depleting run on the exercise-induced hepcidin response the following day was previously investigated. Consequently, on the following day, a low carbohydrate trial (consumption of a low carbohydrate meal after a glycogen-depleting run) induced a significantly higher post-exercise level of plasma IL-6 and pre-exercise level of serum hepcidin [27]. Although the present study was designed as an acute exercise design, it is possible that twice a day endurance exercise with low carbohydrate intake may chronically increase the hepcidin level with augmented exercise-induced IL-6. Moreover, hepcidin production is facilitated by fasting [28], a negative energy balance during exercise [29], and short-term low carbohydrate and high fat diet [30]. We could not determine the exercise energy expenditure over the day, and it was likely that the subjects completed the prescribed exercise sessions under a low-energy-availability status. We speculated that the elevated hepcidin level was associated with the carbohydrate shortage and low energy availability, but further determination is still required to clarify this notion.

Contrary to our expectations, no significant change in haptoglobin was found over time. A substantial degree of exercise-induced hemolysis is commonplace after running, with a decrease in haptoglobin. In addition, previous studies reported that lowered haptoglobin was only observed after running, not after cycling exercise. These results suggest that the mechanical stress of the sole during running might trigger the destruction of red blood cells. Hence, hemolysis frequently occurred in runners who training at high-intensity on a daily basis, and it is possible that subjects in the present study observed chronically low haptoglobin levels [31,32].

In a previous study, the serum hepcidin level increased over the day with iron intake (65 mg/day) and without iron intake [33]. Thus, hepcidin showed diurnal variation, being lower in the early morning and increasing in the afternoon. McCormick et al. (2019) showed that exercise-induced hepcidin elevation was greater in the afternoon than in the morning [34]. Therefore, the diurnal variation may also affect the elevated hepcidin level in the afternoon.

In the present study, we observed that the serum iron level was higher after endurance training; this may have been due to increased hemolysis. Short-term dehydration and diet restriction may increase the production of reactive oxygen species [35]. Moreover, exercise-induced oxidative stress facilitates hemolysis [36]. These findings suggest that endurance training under low energy or carbohydrates augments inflammation and hemolysis, resulting in increased hepcidin secretion.

The present study has several limitations. First, we did not prepare a no exercise trial (control trial) or a single bout of exercise trial because we aimed to collect practical data during real training sessions on the track among well-trained long-distance runners. Previous studies reported that hepcidin level was transiently elevated at ~3 h after completion of the exercise, and that it returned to baseline level after 24 h [8]. On the other hand, in our study, serum hepcidin level remained significantly elevated after 24 h over the baseline level following twice a day endurance exercise in female long-distance runners. We speculate that two bouts of endurance exercise within a day would be the reason for the sustained elevation of serum hepcidin level until the following morning, but direct comparison with a single session of the same exercise will be required in future study. Second, we could not determine energy expenditure during each exercise session. Therefore, it is possible that individual differences in energy balance affected the hepcidin response. Third, we were not able to evaluate plasma IL-6 elevation immediately after completing exercise, due to limited points of blood collections. Therefore, the relationship between IL-6 and hepcidin is not still conclusive. However, to our knowledge, this is the first report to provide practical findings that twice-a-day endurance exercise on the track may increase the risk of ID, due to prolonged elevation of the serum hepcidin level, until the following morning. Further studies are required, but it seems that sufficient energy and carbohydrate intake might overcome increased hepcidin elevation.

## 5. Conclusions

In conclusion, twice a day endurance exercise resulted in significant elevation of serum hepcidin level 24 h after completion of the exercise in female long-distance runners. Therefore, athletes who have a high risk of anemia should pay attention to training frequency and nutritional intake in order to maintain optimal iron metabolism.

## Figures and Tables

**Figure 1 nutrients-14-01907-f001:**
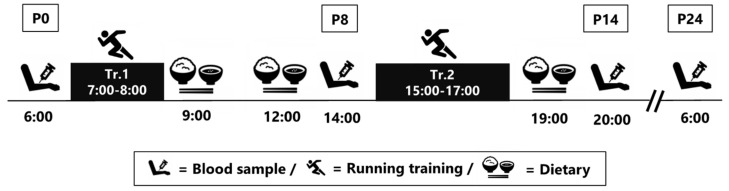
Study protocol.

**Figure 2 nutrients-14-01907-f002:**
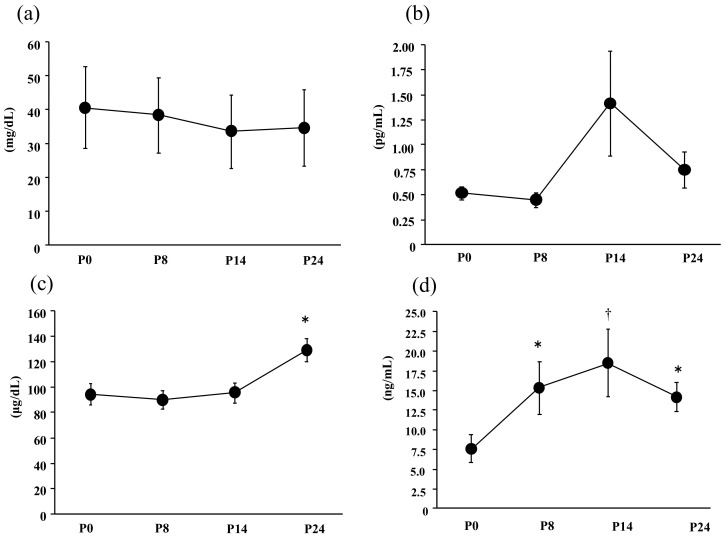
Comparisons of blood variables. (**a**) Serum haptoglobin level, (**b**) plasma IL-6 level, (**c**) serum iron level, and (**d**) serum hepcidin level. The values are means ± SE. * Significant difference from P0 (*p* < 0.05), ^†^ Significant difference from P0 (*p* < 0.01).

**Table 1 nutrients-14-01907-t001:** Comparisons of nutritional intake on the day before exercise and day of exercise.

		On the Day Before Exercise	On the Day of Exercise	*p*-Value
Energy	kcal	2177 ± 55	2153 ± 98	>0.05
Protein	g	101.2 ± 3.3	106.8 ± 5.1	>0.05
%P	%	18.6 ± 0.3	19.9 ± 0.5	>0.05
Fat	g	83.6 ± 2.0	61.2 ± 4.0	<0.001
%F	%	34.6 ± 0.7	25.6 ± 1.3	<0.001
Carbohydrate	g	250.5 ± 7.9	283.4 ± 17.3	>0.05
%C	%	46.8 ± 0.7	54.4 ± 1.7	<0.01
BM	g	5.3 ± 0.2	6.0 ± 0.4	>0.05
Iron	mg	15.6 ± 0.4	15.7 ± 0.6	>0.05

The values are means ± SE. %P: Protein, %F: Fat, %C: Carbohydrate, BM: Body mass.

## Data Availability

The data presented in this study are available on request from the corresponding author. The data are not publicly available due to ethical restrictions.

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
