# Peer review of "Iron Metabolism following Twice a Day Endurance Exercise in Female Long-Distance Runners"

_nutrients, 2022, doi:10.3390/nu14091907_

Round 1

Reviewer 1 Report

This study reports the influence of exercise on systemic iron metabolism. However, iron is present in almost any diet if the individuals do not have any inherited diseases, such as HFE mutation. They will not have a problem absorbing iron from food.  Thus, from this point, this study does not have enough novelty.

Author Response

We thank the reviewers for their valuable comments regarding our manuscript. We have revised the manuscript accordingly. Below is a point list of the reviewer’s comments, along with our responses. We have incorporated the changes in the revised manuscript in underlined text in red.

Reviewer 2 Report

The authors conducted a pilot study investigating the effect of two exercise sessions in one day on markers of iron homeostasis in female long-distance runners. Several comments, however, should be addressed before acceptance for publication:

Methods

Distance and duration of running sessions should be described.

What were the pre-test procedures for InBody testing?

Results

Tables need captions that include definitions of the abbreviations used.

Tables spacing between means, ±, and standard error should not that wide; should be single space between them

Table 1: “P7” is listed not “P8” as described elsewhere in the manuscript.

Table 1: Column for “(mg/dL)” should be wider so it fits on one line.

Table 1: Include all p-values, not “N.S.” for p >0.05.

Discussion

Do the authors have any comment or speculation as to the no change in observed haptogen concentrations?

The authors use two days of self-report dietary logging to infer CHO and, subsequently, glycogen status. Is this sufficient to infer CHO status of these athletes?

The authors seem to emphasize CHO intake and availability perhaps more than is warranted, Line 217 and 223 for example.

             The authors may read the following:

McKay, A., Pyne, D. B., Burke, L. M., & Peeling, P. (2020). Iron Metabolism: Interactions with Energy and Carbohydrate Availability. Nutrients, 12(12), 3692. https://doi.org/10.3390/nu12123692

From the above text:

“Therefore, it appears that there is no additional benefit to iron regulation from increasing CHO intake to very high levels..”

“To date, chronic investigations of CHO restriction (i.e., ketogenic LCHF diets) have not shown clear evidence of negative effects on either iron status or iron regulation”

The study is a pilot study; however, the lack of control group should be mentioned as a limitation of the study.

Other comments:

Line 45: “In reality” might be replaced with “In practice”.

Line 47 – 49: The study referenced is not actually mentioned in this sentence. It is unclear what “the second bout…” is referring to as there was no first bout described. Sentence might be rewritten as “A second bout…”

Line 51: Reference in different style.

Line 62: “Realistic” might be replaced with “practical” or “in-field”.

Line 70: “Dedicated” might be replaced with “exclusive”.

Line 127: Kgs are unit of mass, not weight.

Line 152: “Until” should be replaced with “through”.

Line 161 – 163: The authors should state “These results may be explained by low muscle glycogen…”. Muscle glycogen was not measured in the study, so you can only speculate. Additionally, other exercise factors increase IL-6 such as muscle damage. It is possible two training bouts in one day resulted in muscle damage which may also lead to elevated IL-6. Although, there is a relationship between muscle glycogen concentrations and IL-6, as the authors have described, it is not sole instigator for exercise-induced IL-6. The authors should acknowledge this particularly considering subjects performed two bouts in one day and the authors did not measure muscle glycogen nor did they include a CHO-restricted/limited condition to compare to a CHO-rich condition.

Line 168 – 170 and 177 – 178: The distance of the subjects’ running needs to be described to justify the 8 – 12 g/kg CHO for high-intensity or high-volume training. Subjects consumed a reported 5.3 – 6.0 g/kg CHO, however, without the training distance/volume it is unclear if this is insufficient or not.

Line 196: Would an increase in hemolysis result in an increase in haptogen?

Author Response

(The authors gave the same response as above.)

Reviewer 3 Report

This pilot study examines iron metabolism after twice a day endurance exercise in female long-distance runners. The study is quite intriguing with the appropriate and thorough methods. Statistical analysis was decently conducted; however, the presentation of the results must be improved. The manuscript could be written better, whereas some aspects of the Introduction and Discussion can be significantly improved. Please see some of the comments and raised issues below.

General comments:

Title: I don’t understand why this is a pilot study? There are 13 runners in total. This is a good sample.

Abstract: Authors mention that “Serum hepcidin levels were significantly elevated (compared to P0) until the 21 following morning (P24)” as a result. Then they conclude pretty much the same “These findings suggest that twice a day endurance exercise caused sustained elevation of serum hepcidin level until the next day in female long-distance 24 runners.” Please add a more meaningful conclusion.

Introduction:

  1. More rationale is needed. The Introduction is brief, without a review of the previous studies examining a similar phenomenon. The authors mentioned some previous studies at the end of the Introduction; however, further elaboration on these studies is needed.
  2. Since there are previous studies in this field, some hypotheses should be presented.

Methods:

Subjects:

  1. Please indicate by which criteria they are classified as top-class long-distance runners. My suggestion is to add some of their personal bests. For example, all runners had marathon PB under 3 hours.
  2. Please indicate that study was conducted in accordance with Helsinki Declaration.

Results:

  1. Please indicate that the average running distance was a weekly distance (I assume).

Fasting iron parameters in the morning:

  1. Please indicate the level of significance (F value, p-value, and effect size).

Haptoglobin, iron, IL-6, and hepcidin level after exercise:

  1. Results, in general, are visually badly presented. The authors have decent repeated measures here. A figure with columns or a scatter plot with connected lines would be a better choice than the table and textual explanation of the results.
  2. Furthermore, results of ANOVAs are incomplete. You can indicate significant main effects in the text (e.g., F = 0.852; p = 0.245; n2 = 0.01), and mean +- stdev can be presented in the graph (with * and ** indicating significance at p<0.05 and p<0.01, respectfully.

Nutritional intake during the exercise periods:

  1. Table 2 results might stay in the table since there is no statistical significance; however, since the statistics are performed, you need to add the results of the t-value and p-value.
  2. I assume that you used a T-test for paired samples to test differences in Table 2. Please indicate that in the statistical analysis chapter.

Discussion:

  1. Lines 169-179 are more appropriate for the Introduction rather than Discussion. Please use Discussion to explain and elaborate on obtained results, and not to review previous studies

Specific comments:

Line 51: Citation style is different than the rest of the manuscript

Lines 62-64: This part should be elaborated in the main part of the Introduction.

Author Response

(The authors gave the same response as above.)

Round 2

Reviewer 1 Report

Not enough novelty 

Reviewer 3 Report

I am pleased with all the answers and added changes. They are done in a professional and scientific manner. I've only noticed some overlapping of the numbers in Figure 2. Please make sure that this does not end up in the final version of the manuscript.